# Identification of Plasma Proteins as Biomarkers for Mild Cognitive Impairment and Alzheimer’s Disease Using Liquid Chromatography–Tandem Mass Spectrometry

**DOI:** 10.3390/ijms241713064

**Published:** 2023-08-22

**Authors:** Makoto Inoue, Hideaki Suzuki, Kohji Meno, Shan Liu, Tatsumi Korenaga, Kazuhiko Uchida

**Affiliations:** 1Research Division, MCBI, 5-4-2 Toukoudai, Tsukuba 300-2635, Ibaraki, Japanh.suzuki@mcbi.co.jp (H.S.);; 2Clinical Bioinformatics Initiative, Institute for Biomedical Research, MCBI, 5-4-2 Toukoudai, Tsukuba 300-2635, Ibaraki, Japan

**Keywords:** biomarker, dementia prevention, mass spectrometry, multiple reaction monitoring, proteomics

## Abstract

Blood proteins can be used for biomarkers to monitor the progression of cognitive decline, even in the early stages of disease. In this study, we developed a liquid chromatography–tandem mass spectrometry (LC-MS/MS)-based blood test to identify plasma proteins that can be used to detect mild cognitive impairment (MCI) and Alzheimer’s disease (AD). Using this system, we quantified plasma proteins using isotope-labeled synthetic peptides. A total of 192 patients, including 63 with AD, 71 with MCI, and 58 non-demented controls (NDCs), were analyzed. Multinomial regression and receiver operating characteristic (ROC) analyses were performed to identify specific combinations of plasma protein panels that could differentiate among NDCs, those with MCI, and those with AD. We identified eight plasma protein biomarker candidates that can be used to distinguish between MCI and AD. These biomarkers were associated with coagulation pathways, innate immunity, lipid metabolism, and nutrition. The clinical potential to differentiate cognitive impairment from NDC was assessed using area under the curve values from ROC analysis, which yielded values of 0.83 for males and 0.71 for females. This LC-MS-based plasma protein panel allows the pathophysiology of AD to be followed through detection of cognitive decline and disease progression markers.

## 1. Introduction

Late-onset Alzheimer’s disease (AD), which develops in old age, interferes with daily life through cognitive decline in affected people aged ≥65 years. The pathophysiology of AD is a continuum, progressing from the preclinical phase, in which there are no clinical symptoms, to mild cognitive impairment (MCI) and impairment of daily life [1], and then to dementia.

Clinical interventions can prevent the development of cancer in high-risk individuals if properly diagnosed in a timely manner. Identification of effective diagnostic biomarkers that enable timely and accurate diagnosis would greatly improve the ability to prevent further cognitive decline. Therefore, early detection of MCI and dementia and prevention of dementia through early therapeutic intervention are crucial. Tests for early detection include face-to-face neuropsychological testing, brain imaging, such as magnetic resonance imaging (MRI), and spinal fluid and blood tests. All these tests should be performed during routine health checkups if time and money can be spent; however, blood tests, which are simple and easy to perform, are more widely accessible.

Most of the blood proteins identified as disease biomarkers in previous studies were either disease-related proteins/peptides, such as Ab and phosphorylated tau (P-tau), or proteins associated with neurodegeneration. Focusing on Ab sequestration, we performed proteomics and examined the levels of three blood proteins, apolipoprotein A1 (ApoA1), transthyretin (TTR), and complement protein C3 (C3), and found that their blood levels were decreased in MCI and AD in a multicenter clinical study [2,3].

Recent studies have shown that early intervention in the preclinical stage of MCI is effective in preventing dementia [4]. Efforts to prevent dementia are crucial to reducing the burden of dementia worldwide. However, adequate cognitive functional tests that can be administered in the preclinical stages of MCI are not available currently. Diabetes and hypertension have been shown to be risk factors for the development of AD, and controlling these lifestyle-related diseases from middle age may lead to additional dementia-prevention approaches [5,6]. Blood tests are a potential tool to assess the risk of cognitive impairment and/or evaluate the effectiveness of preventive trials. These drawbacks underscore the need for quantitative proteomic platforms for biomarker discovery of cognitive impairment.

Numerous studies have reported on blood- and CSF-based biomarker discovery for Alzheimer’s disease [2,3,7,8,9,10,11]. The ‘non-biased omics’ strategy encompasses both metabolomic and proteomic profiling using liquid chromatography–tandem mass spectrometry (LC-MS/MS). In the ‘focused proteomics’ approach, multiple immuno-assays are employed to identify plasma biomarker protein profiles associated with specific biological functions in AD pathogenesis. However, there are limited reports addressing plasma protein biomarker profiles in relation to dementia prevention.

In this study, we established a workflow for plasma protein biomarker discovery and used it to identify a plasma protein panel for screening and early detection of cognitive impairment to prevent dementia.

## 2. Results

### 2.1. A Quantitative Identification-Based Proteomic Biomarker Discovery Platform

From a large pool of over 1000 cases, we selected 192 participants, comprising 63 patients with AD, 71 patients with MCI, and 58 non-demented controls (NDCs), based on Alzheimer’s Disease Neuroimaging Initiative (ADNI) criteria. Participants were categorized based on MMSE-J, CDR, and WMS-R LM-II scores, along with clinical assessments and diagnoses by physicians (Table 1).

The initial step of biomarker screening involved determining the relative protein concentrations in plasma by calculating the peak area ratio over the average peak area. Plasma samples from these participants were pretreated, and plasma proteins were digested using trypsin. The resulting tryptic digests were analyzed using an LC-MS/MS system (LCMS-8060; Shimadzu, Kyoto, Japan) (Figure 1). We further refined the selection process using synthetic peptides for precise quantification. At each stage, multinomial regression and receiver operating characteristic (ROC) analyses were employed to evaluate the clinical performance of biomarker candidates.

### 2.2. Semi-Qualitative Analysis for Screening Biomarker Protein Candidates

We successfully identified 45 plasma proteins using a modified version of the method described by Kuzyk et al. [7]. To assess their potential as biomarkers, binary logistic analyses were performed individually or in combination with the two proteins to distinguish between NDCs and those with MCI and between those with NDC and AD. The performances of these protein biomarker candidates were evaluated using ROC analysis. Among the evaluated biomarker pairs, those with the highest area under the curve (AUC) values in the ROC curves were selected as the most promising candidates. To ensure the reliability of our findings, we conducted four independent experiments, and the results consistently demonstrated reproducibility. Figure 2 indicates the 21 plasma proteins identified as candidate biomarkers of MCI and AD.

### 2.3. Quantitative Analysis for the Identification of a Biomarker Protein Panel

Subsequently, we quantified the selected 21 plasma proteins using a set of synthetic peptides as internal standards for LC-MS/MS measurement (Figure 2). To ensure accurate quantification, the concentration of each candidate biomarker was normalized. Then, binomial LASSO regression analysis was performed to identify biomarker proteins associated with cognitive impairment.

Biomarker proteins were categorized based on their functions. Albumin (Alb) was associated with protein nutrition, apolipoprotein C1 (ApoC1) with lipid metabolism, complement components 3 (C3) and 4 gamma chain (C4G) with innate immunity, alpha-2-antiplasmin (A2AP) and alpha-2-macroglobulin (A2M) with coagulation/fibrinolysis, hemopexin (HPX) with heme detoxification, and alpha-1-glycoprotein (A1BG) with inflammation. By categorizing biomarker proteins according to their functions, we gained insights into the potential biological pathways and processes involved in cognitive impairment.

Changes in the levels of these eight plasma proteins in NDC, MCI, and AD are shown in Figure 3 and Table 2.

### 2.4. Composite Marker Consisting of Eight Plasma Proteins That Differentiated Cognitive Impairment from NDC

To examine the differences among the diagnostic groups, we assessed a composite marker consisting of eight plasma proteins previously identified using binomial LASSO regression. The composite score, calculated based on the concentrations of these eight plasma proteins, showed the potential to discriminate between cognitive impairment, including MCI and AD, and the NDC group.

Figure 4 shows the composite scores of the eight plasma proteins in individuals categorized as NDC, early MCI, late MCI, or AD. The scores significantly increased with disease progression in both males and females. Specifically, in males, significant differences were observed between the NDC and AD groups (*p* = 5.9 × 10^−8^), NDC and late MCI groups (*p* = 0.000014), and NDC and early MCI groups (*p* = 0.017). In females, the scores showed an increasing trend in disease progression, and a significant difference was observed between the NDC and AD groups. The composite scores derived from the eight plasma proteins exhibited a high discriminative ability in differentiating those with AD from NDCs, as demonstrated by the AUC values obtained from ROC analysis (Figure 5). Specifically, in male participants, the AUC value for AD vs. NDC was 0.90 with 77% sensitivity and 89% specificity, indicating strong discriminatory power. In female participants, the AUC value for the same comparison was 0.80 with 66% sensitivity and 81% specificity. Furthermore, when considering cognitive impairment (MCI and AD) in comparison with NDC, the composite scores showed good discriminatory performance. In male participants, the AUC for cognitive impairment versus NDC was 0.83 with 73% sensitivity and 86% specificity, whereas it was 0.71 with 66% sensitivity and 72% specificity in female participants.

## 3. Discussion

The composite marker, consisting of eight biomarker proteins identified in this study, allowed differentiation between participants with cognitive decline from those with normal cognition, and the composite scores correlated with progressive cognitive decline. These plasma proteins are associated with various biological processes, including protein nutrition, lipid metabolism, immunity/inflammation, and coagulation/fibrinolysis. Some of these biomarkers are also linked to dementia risk based on lifestyle factors, particularly in the preclinical stages of cognitive decline (Figure 6). Importantly, these biomarkers may have the potential to serve as risk indicators in midlife and old age, which would be valuable for guiding lifestyle modifications aimed at promoting brain health.

Representative blood-based biomarkers include Ab40/42, P-tau 217, P-tau181, glial fibrillary acidic protein (GFAP), and neurofilament light chain (NfL) [12]. A combination of these biomarkers may be useful not only for diagnosis but also for prognosis of cognitive impairment [12,13]. Dysregulation of gene expression leading to altered transcriptional profile in AD pathogenesis may be helpful to identify biomarker protein for MCI and AD [14,15]. In the current study, we employed ‘non-biased proteomics’ rather than ‘focused proteomics’. Consequently, several proteins implicated in AD pathogenesis, such as Ab and tau, and cytokines like NFkB were not identified as biomarkers for MCI and AD. While LC-MS demonstrates high sensitivity in quantifying pure proteins, low-abundance proteins may be obscured by more abundant proteins in a complex plasma sample. This omission might be attributed to their concentration levels in plasma.

Statins not only lower low-density lipoprotein (LDL) levels but also elevate high-density lipoprotein (HDL) levels [16], and additionally, it may have an effect on the levels of inflammatory proteins in plasma [17]. In this study, some participants, but not all, might be on active medications, including statins. These medications could potentially alter plasma protein levels, leading to false positive or negative results.

Of the eight plasma proteins identified, some of them exhibited altered plasma levels in cases of cognitive impairment and showed an association with the risk of cognitive impairment in longitudinal studies [18,19,20]. We identified APOC1, A2AP, HPX, and A1BG as emerging protein biomarker candidates for MCI and AD. These results hint at a potential link between plasma proteins engaged in coagulation, fibrinolysis, and AD pathogenesis. It is significant to note that the processes of coagulation and fibrinolysis, which are components of the contact system, interplay with the inflammatory cascade, particularly complement activation in AD pathogenesis.

The recent literature suggests a possible association between the contact system and the integrity of capillary vessels in the brain, a crucial element in AD pathology [21,22]. As brain capillaries age, they become susceptible to vascular damage, leading to a compromised blood–brain barrier (BBB). The activation of the coagulation cascade may act as a protective mechanism, preventing toxic agents from infiltrating the brain parenchyma by promoting clot formation. Thus, the protein biomarkers unearthed in our study may provide invaluable insights for the early diagnosis and proactive prevention of AD.

Alb binds to Aβ and plays a role in inhibiting self-aggregation. Approximately 90% of Aβ present in circulating peripheral blood is bound to Alb [23,24]. Alb is involved in Aβ clearance in the brain and peripheral blood [25]. Decreased blood Alb levels have been associated with increased Aβ deposition, as measured by amyloid PET [26]. Additionally, Llewellyn et al. reported an association between low serum Alb levels and risk of cognitive impairment in older individuals [18].

ApoC1 inhibits lipoprotein lipase and cholesteryl ester transfer protein, induces lecithin-cholesterol acyl transferase, and helps regulate blood lipid levels [27]. It acts on lipoprotein receptors and regulates the activities of several enzymes involved in lipid metabolism. This apolipoprotein accounts for approximately 10% of very low-density lipoprotein and 2% of high-density lipoprotein (HDL) protein components. ApoC1 exhibits a modest correlation with HDL and low-density lipoprotein and a relatively strong correlation with total cholesterol [27]. Abnormal ApoC1 levels are associated with cardiovascular diseases, diabetes, and cognitive dysfunction [28]. In senile plaques of individuals with AD, coexistence of ApoC1 and Aβ has been observed [29]. Moreover, ApoC1 is associated with the *APOE* gene, and individuals with the ε4 gene variant of this gene are known to express low levels of ApoC1. Overexpression of ApoC1 in mice has also been shown to result in cognitive dysfunction [29].

Under normal conditions, the immune system eliminates neurotoxic agents, such as Aβ. However, dysregulated immunity can lead to diseases, such as AD, which result from excessive inflammation and tissue damage. Complement components are humoral factors that play central roles in innate immunity. They bind to and activate external and foreign substances, including bacteria, as part of the body’s defense mechanisms. In the brain, complement components are essential for microglial activation and efflux of such substances [30]. Microglia are immune cells in the brain that become active in response to nerve damage. They phagocytose dead cells, release factors that promote repair, and help maintain synaptic connections [31].

With progressive cognitive decline, there is a decrease in plasma C3 protein levels. This reduction may be attributed to the consumption of complement components caused by chronic inflammation, which is believed to be involved in the pathophysiology of AD. Cohort studies, including a study by Rasmussen et al. [20], have reported that low blood C3 levels are associated with increased risk of dementia.

Brain microvascular damage is considered an early stage of AD pathological progression. This damage can lead to leakage through the blood–brain barrier (BBB) and subsequent infiltration of inflammatory cells into the brain parenchyma. When the BBB is compromised, blood components, including blood cells, enter the brain. Hemoglobin present in red blood cells contains iron, which contributes to oxidative stress, inflammation, and subsequent neurological damage.

Hpx, a heme-binding protein, plays a critical role in mitigating hemoglobin toxicity. Plasma levels of Hpx are often elevated in inflammatory diseases and serve as potential indicators of inflammation in cerebral vascular endothelial cells [32,33]. Notably, in AD-affected brains, amyloid deposits and microhemorrhages (hemoglobin deposits) are located in close proximity [34], and amyloid deposition has been shown to correlate with amyloid PET imaging and plasma hemopexin concentration [35].

These observations suggest a link between amyloid deposition, microvascular damage, and hemoglobin-related processes in patients with AD. The presence of amyloid deposits and microhemorrhages at the same site in the AD brain, along with the correlation between amyloid deposition and plasma Hpx concentrations, highlight the potential interplay between these factors in AD pathogenesis. Therefore, monitoring Hpx levels can provide insights into the inflammatory status and cause potential damage to the brain when the BBB is compromised.

Altered levels of A2M in the blood have been observed in large U.S. cohort studies, such as the ADNI and Baltimore Longitudinal Study of Aging [19]. Plasma A2M levels have been found to be elevated in individuals with MCI and AD compared with those with normal cognitive function. Furthermore, high baseline plasma A2M levels have been associated with an increased risk of developing MCI and AD in longitudinal studies.

A2M functions by inhibiting the coagulation reaction through its ability to bind and degrade thrombin [36]. Increased levels of A2M can effectively inhibit coagulation, which is necessary for repairing BBB damage in capillary endothelial cells. However, an excessive increase in A2M levels can delay the repair of vascular endothelial cells, allowing blood components to enter the brain. This influx of blood components can lead to inflammation and nerve cell damage. It has also been reported that plasma A2M levels increase in conjunction with BBB damage. To facilitate repair of endothelial cell damage within capillaries of the BBB, blood coagulation initiates fibrin formation to plug the damaged area. Once the endothelial cells have been repaired, fibrin is rapidly broken down by plasmin, an enzyme responsible for fibrinolysis. During this process, A2AP binds to plasmin, inhibiting its fibrinolytic activity and stabilizing the clot until the vascular endothelial cells are fully repaired. Downregulated A2AP expression may compromise endothelial cell repair. Consequently, fibrin degrades and blood components can enter the brain, leading to inflammation and nerve cell damage. Studies using mouse models of AD have shown that suppression of A2AP expression increases plasmin activity, resulting in enhanced Aβ deposition and inflammation in the brain [37,38].

In individuals with MCI and AD, blood plasmin levels are lower than those in cognitively normal participants. This decrease in plasmin levels is associated with low A2AP levels, reduced blood Aβ42 levels, Aβ42 accumulation in the brain, and hippocampal atrophy (unpublished observations). These findings suggest that dysregulation of A2AP and plasmin activity may contribute to pathological processes underlying MCI and AD, including Aβ deposition, inflammation, and neurodegeneration.

A1BG is a plasma protein that belongs, structurally, to the immunoglobulin superfamily; however, its precise function has not been fully elucidated [39]. However, A1BG has been implicated in inflammatory processes and has emerged as a potential biomarker of AD based on proteomic analysis [40]. Increased levels of A1BG have been identified as a risk factor for the development of AD. Correlations were observed between A1BG levels and fluctuations in C3 and HPX levels. Although the exact role of A1BG in AD and its involvement in inflammatory pathways remain unknown, the association between increased A1BG levels and AD, as well as its connection with C3 and HPX, suggests that A1BG may play a role in the inflammatory processes and pathological mechanisms underlying AD.

The biomarker proteins identified in this study are believed to play a key role in AD pathophysiology. These proteins are associated with functions that involve clearance of brain waste, protection against neurotoxicity, and maintenance of healthy brain vessels. To promote and maintain brain health, it is essential to adopt a balanced diet that includes the appropriate proportions of proteins, fats, and carbohydrates. This nutritional balance is crucial to ensure a healthy and adaptable brain, thereby supporting cognitive function and preserving overall brain health.

The present findings indicate that composite scores derived from the eight plasma proteins can be used to effectively differentiate those with AD from NDCs and identify cognitive impairment when compared to NDCs. The high AUC values obtained from the ROC analysis support the utility of this marker as a tool for the early detection and monitoring of cognitive decline.

Although further studies are required to assess the clinical utility of blood tests in evaluating cognitive decline, the identified biomarker protein panel holds promise for the early detection of changes associated with lifestyle diseases that act as risk factors for dementia. This panel, consisting of specific plasma proteins, is a potential tool to identify early indicators of cognitive impairment and monitor disease progression.

## 4. Materials and Methods

### 4.1. Reagents

Ammonium bicarbonate was purchased from Kishida Chemical Co., Ltd., Osaka, Japan, CHAPS from Nacalai Tesque Inc., Kyoto, Japan, and sequence-grade modified trypsin from Promega, Madison, WI, USA. Bovine serum albumin, guanidine hydrochloride, formic acid, and acetonitrile were purchased from FUJIFILM Wako Pure Chemical Corporation, Osaka, Japan, while n-Octyl-β-D-glucoside was purchased from Dojindo, Tokyo, Japan.

### 4.2. Preparation of Peptide Solutions

Synthetic peptides were purchased from Eurofins Genomics Inc., Tokyo, Japan. Each peptide powder was dissolved in GuOG (6M guanidine hydrochloride, 0.1% n-Octyl-β-D-glucoside) as a 1.5 mg/mL solution and were quantified using LC-UV chromatogram under predicted molar extinction coefficient at λ = 214 nm [41]. Peptide mixtures for either the LC-MS reference or internal standard were prepared by mixing peptide solutions, which were then diluted in peptide diluent (0.69% formic acid, 7.25 mM ammonium bicarbonate, 30 mM guanidine hydrochloride, 0.12% CHAPS).

### 4.3. Human Participants

Participants were selected according to the following inclusion criteria: no history of treatment for any serious disease (cancer, myocardial infarction, stroke) within 5 years; and no dementia treatment (Aricept, Reminyl, Rivastatin, or Memary). The exclusion criteria were as follows: history of therapeutic inpatient surgery due to a stroke, subarachnoid hemorrhage, or other head injury; and the use of medications that may affect this study (e.g., antipsychotics, anxiolytics, antidepressants).

Participants with any psychiatric illness according to the Diagnostic and Statistical Manual of Mental Disorders, Fourth Edition (DSM-V) criteria were excluded from the study. We defined NDCs as participants with MMSE-J scores of 30, without atrophy of the hippocampus on MRI scans and without obvious decreased rCBF on SPECT imaging. MCI was defined as an MMSE-J score of 24–27, atrophy, and decreased rCBF, according to Peterson’s criteria. AD was defined according to the DSM-V criteria: participants with AD had an MMSE-J score below 24, hippocampal atrophy, and decreased rCBF. For LC-MS analysis, we selected plasma samples from 192 participants who met the ADNI criteria. The protocol for blood sample collection was approved by the ethics committee of each institute. Prior to inclusion in the study, written informed consent was obtained from each participant, and all samples were rendered anonymous before being sent to the laboratory for analysis.

### 4.4. Plasma Sample Preparation and Protein Digestion

Blood was collected in 5 mL EDTA-2Na tubes (Venoject-II Tube; Terumo, Tokyo, Japan). The tubes were gently mixed by inverting 8–10 times, followed by centrifugation at 1200× *g* at 20 °C for 20 min. Plasma samples were prepared as 40 μL aliquots and stored at –80 °C until LC-MS analysis. All serum samples were frozen and thawed twice.

To prepare the plasma protein digests, 3 μL of EDTA plasma was diluted in 22.5 μL of plasma diluent (25 mM AB, 3.3% CHAPS, 33.3 μg/mL BSA), followed by heat denaturation at 99.9 °C for 5 min in a thermal cycler (GeneAmp PCR system 9700; Thermo Fisher Scientific, Waltham, MA, USA). An 8.5 mL volume of the heat-denatured sample was digested by adding 43.5 mL activated trypsin [0.5 mg/mL trypsin: 25 mM AB = 2:45 (*v*/*v*)], followed by 16 h incubation at 37 °C. To terminate the reaction, 140 mL of 1% formic acid was added. For the internal standard, 10 μL of stable-isotope-labeled peptide mixture was added.

### 4.5. LC-MS Analysis

All LC-MS analyses were carried out using a triple quadrupole mass spectrometer LCMS-8060 system (Shimadzu, Kyoto, Japan) with an Aeris PEPTIDE XB-C18 column (2.6 μm × 50 × 2.1 mm) (Phenomenex, Torrance, CA, USA). All chromatographic analyses were performed by injection of 1 μL protein digest, followed by a linear gradient of solvent A (0.1% formic acid, 2% acetonitrile) and solvent B (0.1% formic acid, 90% acetonitrile) at a flow rate of 0.2 mL/min.

For quantitative analyses, peptides were separated by an 8.1 min linear gradient of solvent B from 0% to 35%, followed by a 1.4 min linear gradient of solvent B from 35% to 95%. After each separation step, the column was washed with 95% solvent B for 1.5 min and re-equilibrated with 0% solvent B for 4 min. To quantify human plasma proteins, the target peptides and their pairs of precursor/product ions were prepared as described by Kuzyk et al. [42]. Chromatographic data were processed using LabSolutions (Shimadzu) to obtain either peak area or concentration values.

### 4.6. Statistical Analysis

ROC analysis, C-statistics, and Mann–Whitney U test were performed using GNU R “https://www.r-project.org/ (12 February 2021). “The point closest to the upper-left corner of the ROC curve yielded optimal sensitivity and specificity values. Multinomial logistic regression analyses and graphical representations were performed using the R packages of “nnet” and “ggplot”, respectively. *p* values of ≤0.05 were defined as significant.

## Figures and Tables

**Figure 1 ijms-24-13064-f001:**
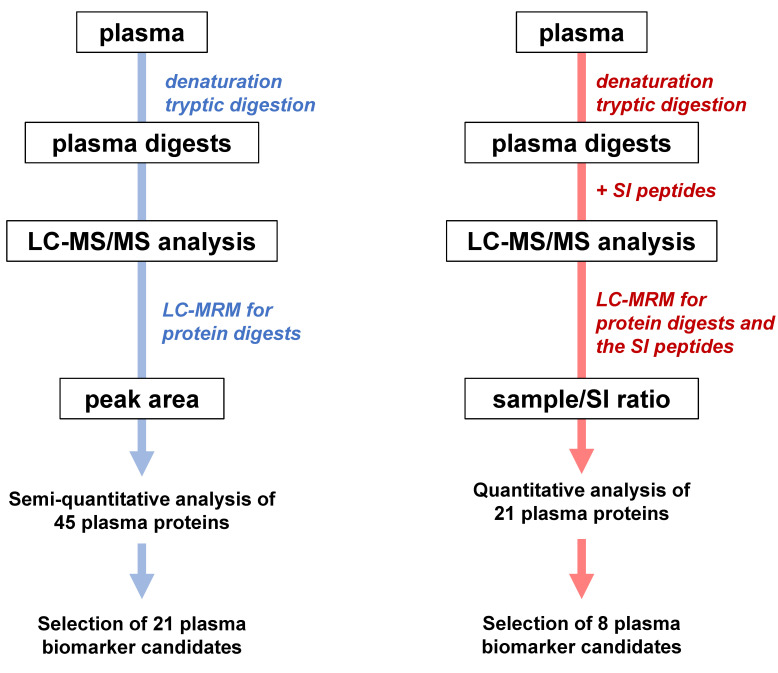
**Strategy for identification and quantitative determination of plasma biomarkers of cognitive impairment.** We assessed the value of these proteins in distinguishing between different groups, namely NDC vs. MCI or NDC vs. AD. ROC analyses were then conducted to evaluate the performance of protein candidates or protein combinations as biomarkers. Quantitative analysis was performed to identify a biomarker protein panel. The relative plasma protein concentrations were determined using a quantitative approach. Ratios of peak area over average peak area were used to determine the relative abundance of proteins in plasma samples (blue). This analysis allowed us to assess the quantitative differences in protein levels among the study groups. Logistic regression analysis was performed for all 45 pairs of plasma proteins. A total of 21 out of 45 plasma biomarker proteins were selected using binary LASSO analyses and their performance was evaluated with ROC analysis. We created sets of two plasma biomarker proteins, and the clinical performance of each combination was tested using ROC analysis. Proteins with the highest AUC values were selected (**left** panel). In the selection process of 8 out of the 21 plasma biomarker proteins, we created a set of synthetic peptides to serve as internal standards for all 21 plasma proteins in LC-MS/MS measurements (red). Using multinomial logistic regression analysis, we investigated combinations of up to 6 plasma proteins selected from a pool of 21 protein biomarker candidates, and we also factored in their biophysiological significance in AD pathogenesis (**right** panel).

**Figure 2 ijms-24-13064-f002:**
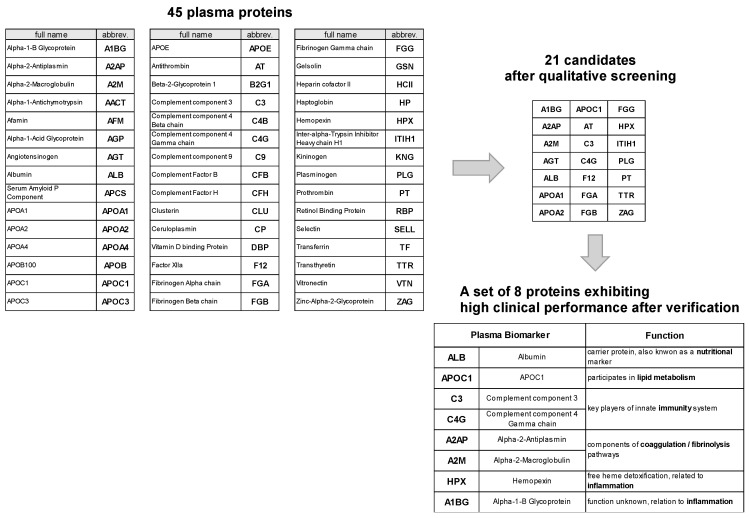
**Plasma biomarker panels for differentiating NDCs from those with MCI and/or AD.** After qualitative and quantitative verification of 45 plasma biomarker candidates, 8 promising biomarker proteins were identified. These eight biomarker proteins were related to protein nutrition, lipid metabolism, innate immunity, coagulation/fibrinolysis, heme detoxification, and inflammation.

**Figure 3 ijms-24-13064-f003:**
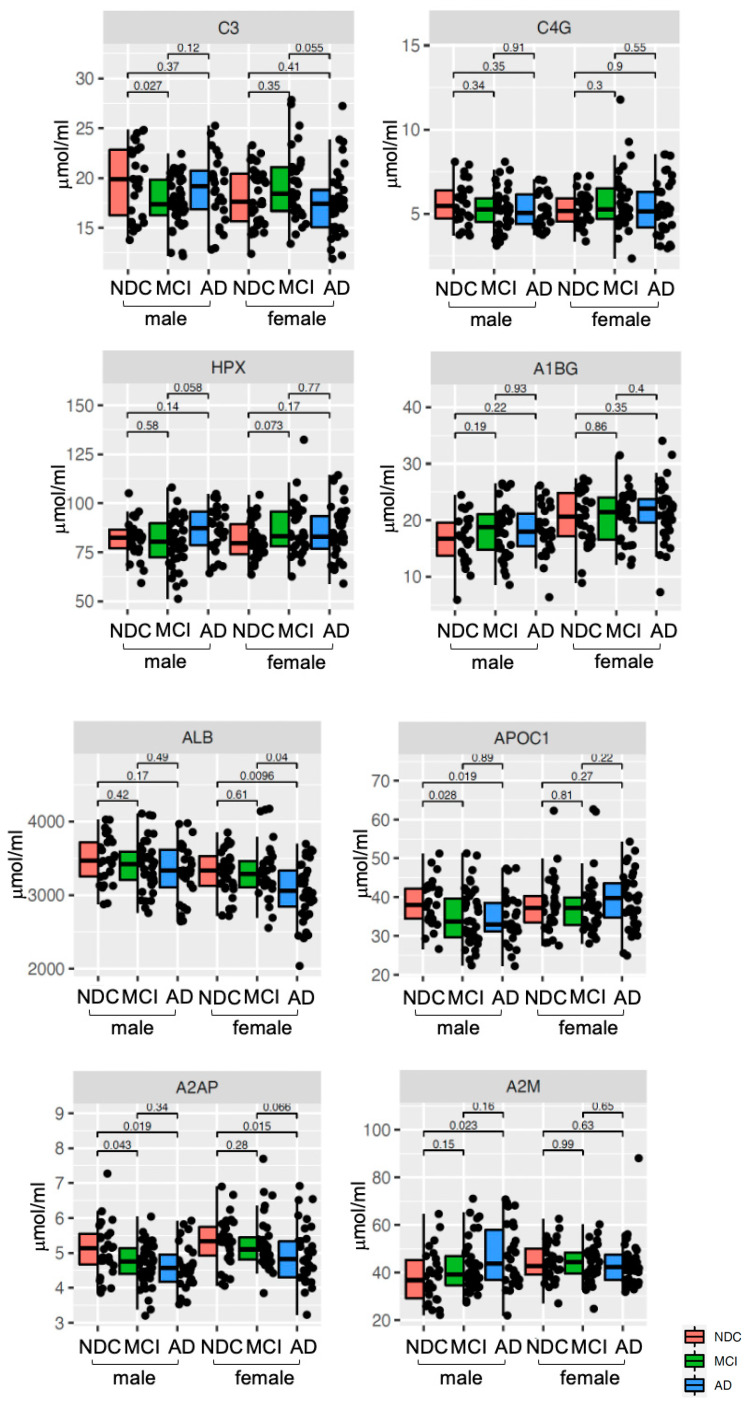
**Plasma levels of eight biomarker proteins in NDC, MCI, and AD.** Plasma levels of the eight biomarker proteins for male and female participants are presented for NDC (red box), MCI (green box), and AD (blue box). Within the boxplot, the bold solid bars denote the mean abundance of each group, while the error bars delineate ± 1.5 SD. Significant variations among the three groups are highlighted, as determined by the Kruskal–Wallis U test. *p* values are indicated.

**Figure 4 ijms-24-13064-f004:**
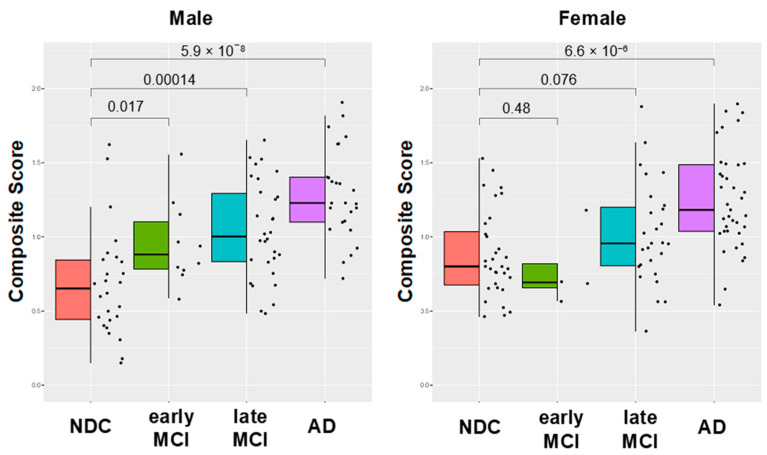
**Composite scores calculated from concentrations of eight plasma biomarkers in disease progression.** Composite scores designed using logistic models for discrimination between either NDCs and those with MCI or NDCs and those with AD exhibited a correlation with severity of cognitive decline based on clinical manifestations in both male and female participants.

**Figure 5 ijms-24-13064-f005:**
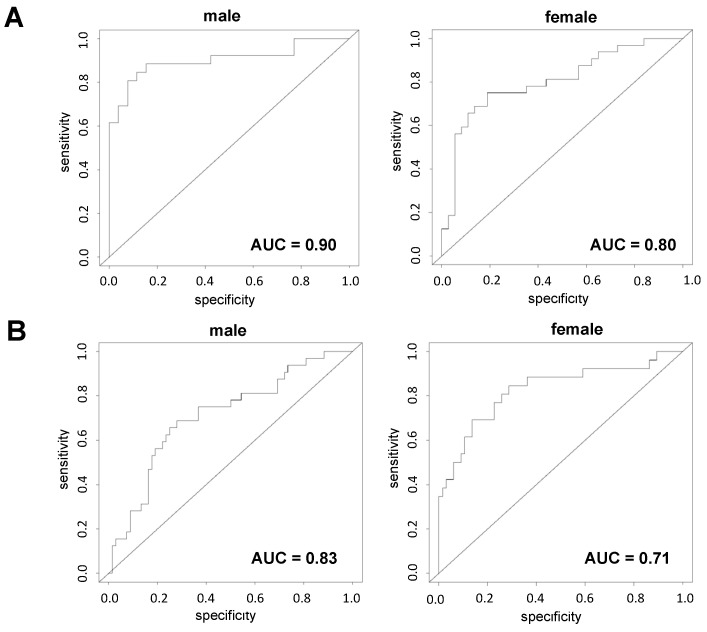
**Clinical potential of composite score consisting of eight plasma proteins.** ROC analysis of composite score consisting of eight plasma proteins in male and female participants evaluated using coefficients of the logistic models was performed. In the model discriminating between NDCs and those with AD, the AUC values of ROC analysis were 0.90 and 0.80 for male and female participants, respectively (**A**). In the model discriminating between NDCs and cognitive impairment (MCI and AD), the AUC values from the ROC analysis were 0.83 and 0.71 for male and female participants, respectively (**B**).

**Figure 6 ijms-24-13064-f006:**
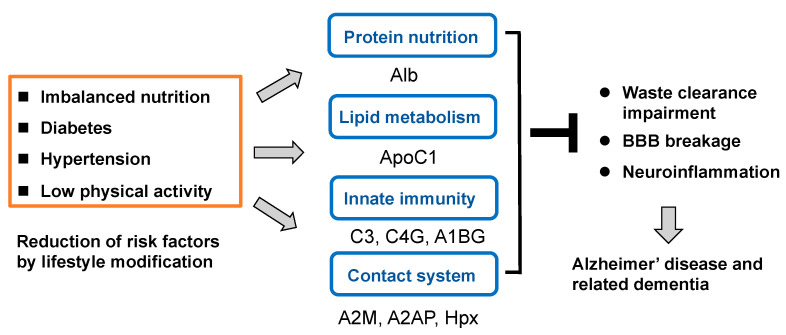
**Plasma biomarker panel for dementia prevention.** Plasma biomarkers found in present study were categorized into four functional groups. Alb was associated with protein nutrition, ApoC1 with lipid metabolism, C3 and C4G with innate immunity, A2AP and A2M with coagulation/fibrinolysis, HPX with heme detoxification, and A1BG with inflammation.

**Table 1 ijms-24-13064-t001:** Clinical characteristics of participants.

	NDC	MCI	AD
	(*n* = 58)	(*n* = 71)	(*n* = 63)
Age	66.7 ± 7.2 ^1^	72.6 ± 6.4	75.8 ± 7.3
Male/female	26/32	40/31	26/37
MMSE-J ^2^	29.3 ± 0.7	26.8 ± 1.7	18.4 ± 4.4
CDR ^3^	0.0 ± 0.0	0.5 ± 0.0	1.3 ± 0.5
WMS-R LM-II ^4^	11.6 ± 3.2	3.6 ± 3.1	0.5 ± 0.9

^1^ Mean ± SD. ^2^ Mini Mental State Examination-Japanese version. ^3^ CDR: Clinical Dementia Rating. ^4^ Wechsler Memory Scale-Revised Logical Memory II.

**Table 2 ijms-24-13064-t002:** Alterations in the levels of the eight plasma biomarker proteins in MCI and AD compared to NDC.

	Biomarker Protein	MCI	AD
Male	C3	↓↓ ^1^	→ ^2^
	C4G	→	→
	HPX	→	→
	A1BG	→	→
	ALB	→	→
	APOC1	↓↓	↓↓
	A2AP	↓↓	↓↓
	A2M	→	↑↑ ^3^
Female	C3	→	→
	C4G	→	→
	HPX	↑ ^4^	→
	A1BG	→	→
	ALB	→	↓↓
	APOC1	→	→
	A2AP	→	↓↓
	A2M	→	→

^1^ significant decrease, *p* ≤ 0.05; ^2^ no change, *p* > 0.1; ^3^ significant increase, *p* ≤ 0.05; ^4 ^increase trend, *p* ≤ 0.10.

## Data Availability

The data presented in this study are available on request from the corresponding author. The data are not publicly available due to ethical issues.

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
