# Peer review of "Identification of Plasma Proteins as Biomarkers for Mild Cognitive Impairment and Alzheimer’s Disease Using Liquid Chromatography–Tandem Mass Spectrometry"

_ijms, 2023, doi:10.3390/ijms241713064_

Round 1

Reviewer 1 Report

This work by Inoue et al is quite important as it reveals the presence of a subset of AD-associated proteins. The authors identified eight plasma proteins that could potentially serve as biomarkers of MCI and AD. These findings open a major new area of work central to understanding the major risk factors for MCI and AD and will make possible a more systematic approach across the field in terms of putting protein-based work in a much needed broad context.  I have some concerns which if addressed can improve the paper. 

1. Was this a fasting blood sample? In the methods section there is no mention of whether the MCI and/or AD patients were on active medications (statins, anti-diabetic, anti-inflammatories etc)? If these medications have an effect on the DNA/protein patterns it could lead to false positive/negative results.

2. The authors identified 45 plasma proteins from which 21 were found to be relevant of which 8 were concluded to be specific for MCI and AD. This needs to be clarified. What was the rationale to narrow down the 45 to 21 and then to 8? On what basis were the 8 proteins identified to be important in the AD process?

3. On page 5, the authors show composite scores of the eight plasma proteins. The scores significantly increased with disease progression in both males and females. However, it is not clear which proteins increased or decreased in the disease process? A table comparing the protein patterns (which ones are decreased or increased) in the controls, MCI and AD patients would be helpful.

4. It is not clear if the pattern of the 8 proteins differed between controls, MCI and AD patients? A representative protein profile like blots or immunohistochemistry would be very helpful to distinguish these differences?

5. Are these 8 proteins present in the brain? If yes, are they present in specific areas of the brain?

6. Why did the authors fail to see presence of APP, Tau, Sirtuins, NFkB, MAP kinase-activating death domain (MADD), activity-dependent neuroprotective protein (ADNP), copper metabolism gene MURR1 (mouse U2af1-rs1region1)-domain 6 (COMMD6), Klotho proteins—which have all been implicated in AD pathogenesis?

7. Do these 8 proteins have differential cellular localization? Within the brain, where are they predominantly localized?

8. It would be appropriate if the authors look at any literature results of assessing proteins in MCI or AD and then comment on whether any of these previously reported proteins are included in the list reported here (for example: see the literature by Simpson, JE et al, Neurobiol Aging, 2011; Theendakara et al, J. Neurosci, 2016; Anuschka Silva-Spínola et al, 2023, Eur J Neurol).

Author Response

We sincerely thank the Reviewers for their valuable insights and comments, which have significantly enhanced the quality of our manuscript. We have diligently addressed all the concerns raised. The Reviewers' insights were pivotal, and we are grateful for the chance to clarify our research objectives and findings. As detailed below, we have reviewed and incorporated the suggested changes, and addition of  several references based on the reviewers' comments.

Please see the attached files.

Reviewer 2 Report

ijms-2529500  Article Identification of plasma proteins as biomarkers for mild cognitive impairment and Alzheimer's disease using liquid chromatography-tandem mass spectrometry

Blood proteins can be used for biomarkers for monitoring the progression of cognitive decline, even in the early stages of disease. In this study, the authors developed a liquid chromatography-tandem mass spectrometry (LC-MS/MS)-based method to identify plasma proteins that can be used to detect mild cognitive impairment (MCI) and Alzheimer's disease (AD). Using this system, the authors quantified plasma proteins using isotope-labeled synthetic peptides.

·         What is the central question addressed by the research?

In this study, the authors developed a liquid chromatography-tandem mass spectrometry (LC-MS/MS)-based blood test to identify plasma proteins that can be used to detect mild cognitive impairment (MCI) and Alzheimer's disease (AD). A total of 192 patients, including 63 with AD, 71 with MCI, and 58 non-demented controls (NDCs), were analyzed. Multinomial regression and receiver operating characteristic (ROC) analyses were performed to identify specific combinations of plasma protein panels that could differentiate among NDCs, those with MCI, and those with AD. The authors identified eight plasma protein biomarker candidates that can be used to distinguish between MCI and AD. These biomarkers were associated with coagulation pathways, innate immunity, lipid metabolism, and nutrition.

·         Do you consider the topic original or relevant in the field? Does it address a specific gap in the field?

The authors have not done anything new here as multiple articles are available for this kind of study.

Liu, Y., Li, N., Zhou, L., Li, Q. & Li, W.. Plasma metabolic profiling of mild cognitive impairment and Alzheimer's disease using liquid chromatography/mass spectrometry.. Central nervous system agents in medicinal chemistry, 2014.

Marksteiner, J.. Five out of 16 plasma signaling proteins are enhanced in plasma of patients with mild cognitive impairment and Alzheimer's disease. Neurobiology of Aging 2015.

·         What does it add to the subject area compared with other published material?

No, no biomarkers identified are very specific for the cognitive decline.

·         What specific improvements should the authors consider regarding the methodology? What further controls should be considered?

Also there is no consistency to the literature data and exisitng methods

·         Are the conclusions consistent with the evidence and arguments, and do they address the main question?

No comments

·         Are the references appropriate?

The above references are missing

·         Please include any additional comments on the tables and figures.

No comments 

Author Response

We sincerely thank the Reviewers for their valuable insights and comments, which have significantly enhanced the quality of our manuscript. We have diligently addressed all the concerns raised. The Reviewers' insights were pivotal, and we are grateful for the chance to clarify our research objectives and findings. As detailed below, we have reviewed and incorporated the suggested changes based on the reviewers' comments.

Please see the attached files.

Round 2

Reviewer 2 Report

Are the concerns are sucessfully revised and answered by the authors.